# “Corp-Osa-Mente”, a Combined Psychosocial–Neuropsychological Intervention for Adolescents and Young Adults with Fragile X Syndrome: An Explorative Study

**DOI:** 10.3390/brainsci13020277

**Published:** 2023-02-07

**Authors:** Federica Alice Maria Montanaro, Paolo Alfieri, Stefano Vicari

**Affiliations:** 1Child and Adolescent Neuropsychiatry Unit, Department of Neuroscience, Bambino Gesù Children’s Hospital, IRCCS, 00165 Rome, Italy; 2Department of Education, Psychology, Communication, University of Bari Aldo Moro, 70121 Bari, Italy; 3Department of Life Sciences and Public Health, Università Cattolica Del Sacro Cuore, 00168 Rome, Italy

**Keywords:** Fragile X Syndrome, *FMR1* gene, Fragile X messenger ribonucleoprotein 1, intellectual disability, cognitive-behavioral therapy, occupational therapy, cognitive training

## Abstract

Fragile X Syndrome is the most known inherited form of intellectual disability due to an expansion in the full mutation range (>200 CGG repeats) of the promoter region of the *FMR1* gene located on X chromosomes leading to gene silencing. Despite clear knowledge of the cognitive-behavioral phenotype of FXS and the necessity of tailored interventions, empirical research on the effectiveness of behavioral treatments among patients with FXS is still lacking, with studies on adolescents and young adults even more insufficient. Here we present “Corposamente”, a combined psychosocial–neuropsychological intervention conducted with a group of ten adolescents/young adults with FXS, who are non-ASD and without significant behavioral problems. In total, 20 sessions were performed, alternating between online and face-to-face meetings. At the end of the intervention, participants, family members and participants’ educators anonymously completed a survey that was designed around key areas of improvement as well as treatment satisfaction. The survey results indicated that participants improved mostly in their ability to cope with negative emotions and that occupational intervention was considered the most effective technique both from families and participants. Our exploratory study suggests that group therapy for the management of the FXS cognitive-behavioral phenotype may be a promising approach to continue to pursue, mostly in adolescence when the environmental demands increase.

## 1. Introduction

Fragile X Syndrome (FXS) is an X-linked neurodevelopmental disorder, averagely affecting as many as 1 in 4000 males and 1 in 8000 females [1]. It is a rare inherited condition, also known as Martin–Bell syndrome, taking the names of the scientists who first described the syndrome in 1943. However, the molecular basis of FXS were only first described in 1991 when the *FMR1* (Fragile X Mental Retardation 1, renamed Fragile X messenger ribonucleoprotein 1) gene located at Xq27.3, whose “full mutation” causes the syndrome, was isolated for the first time [2]. Today, it is well-known that FXS is determined by the silencing of the *FMR1* gene that encodes for FMRP (Fragile X Mental Retardation Protein) protein, which in turn has the function of controlling the translation of specific messengers that are involved in the maturation and function of neuronal synapses [3]. More specifically, FXS is caused by the large expansion of the trinucleotide CGG in the untranslated portion 5 of this gene, which tends to change in amplitude from one generation to the next one [4]. The *FMR1* gene is located on each X chromosome, in which the CGG triplet repeats a certain number of times; when the expansion of the nitrogenous bases exceeds 200 repetitions, methylation of the promoter site is observed, resulting in the silencing of the gene, in the total or partial absence of the FMRP protein for which it encodes, and thus in the manifestation of FXS. In the non-affected population, the CGG repeat number is between 5 and 45. Premutation occurs when the trinucleotide repeats between 55 and 200 times. In such circumstances, the mutation is not complete and the FMRP protein is not totally absent, so there is no full manifestation of the syndrome. However, other clinical conditions such as primary ovarian insufficiency (*FXPOI*), fragile X-associated neuropsychiatric disorders (*FXAND*) and fragile X-associated tremor/ataxia syndrome (*FXTAS*) can be observed [5].

The manifestations of FXS are variable, they change based on sex, age, *FMR1* level of methylation and environmental influences, although some specific features, both on a physical and on a neurobehavioral level, can be identified. The physical characteristics include prominent ears, a long face, flat feet and macro-orchidism in males [6]. Furthermore, FXS is the most common inherited cause of intellectual disability (ID), with 90% of males and around the 30–50% of females exhibiting IQ scores in this range (QI < 70) [3]. Approximately 60% of individuals with FXS have a concurrent diagnosis of ASD (autism spectrum disorder), while ADHD (attention deficit and hyperactivity disorder) affects around 70% of FXS people [7]. The cognitive phenotype of FXS is characterized by prominent executive function (EF) problems, including issues with working memory (WM), cognitive flexibility and selective and divided attention [8], learning disabilities and language problems [9]. The neurobehavioral features involve anxiety, irritability, shyness (especially among women), repetitive behaviors and socio-pragmatic deficits. Furthermore, individuals with full *FMR1* mutations, mostly female ones, may exhibit strategic and theatrical behaviors to gain other people’s attention [10]. Adaptive behavior deficits have been reported too, with a decline or a plateau with growth, especially during adolescence when the environmental demands increase [11]. Adaptive profile seems to be inhomogeneous and characterized by relative strengths in domestic and daily living skills, and weaknesses in socialization and communication abilities [12,13]. 

Despite the clear knowledge about the cognitive-behavioral phenotype of FXS and the need of structured behavioral interventions, empirical studies on the effectiveness of behavioral treatments among patients with FXS are still lacking [10]. Indeed, even though it has been observed that challenging behaviors and socio-relational deficits have the greatest impact on patients and their families (more than ID) [14] and that educational programs are associated with better behavioral outcomes [15], to date there is still limited evidence for the basis of FXS-tailored behavioral interventions. For instance, the systematic review conducted by Moskowitz et al. [16] suggested that individuals with FXS might exhibit improvements after behavioral interventions, pointing out that a behavioral analytic approach to FXS intervention should be pursued. However, the interventions included in the review involved a variety of strategies, often with just one participant for each technique, making it difficult to draw conclusions about the effectiveness of specific treatments. Furthermore, to date most studies have focused on toddlers and children, with studies on adolescents and young adults being even more insufficient. Consequently, while recommendations for medications in FXS are already offered [17], there are no systematic guidelines for behavioral treatments, and for this reason psychologists generally still rely on their clinical experience [18]. 

Considering the above, here we present “Corposamente” (CoM), a behavioral intervention conducted with a group of ten adolescents/young adults with FXS in Apulian, a region of southern Italy. We are aware of the strong methodological limitations of our work. On the other hand, we believe that describing the intervention, illustrating parents and patients’ post-treatment observations and sharing the qualitative results may be important for the research community, in order to replicate the study by providing standardized data and to inspire other colleagues, therefore trying to win the gap between clinical practice and research.

In the following paragraphs, CoM’s methodology and techniques are described. Furthermore, the qualitative findings from a survey that was designed around key areas of improvement that was completed by participants, family members and participants’ educators are provided. 

## 2. Materials and Methods

### 2.1. Participants 

The intervention was performed with ten participants (M:F = 7:3) with *FMR1* full mutation, as determined by DNA testing (Methylation PCR method test). Individuals were recruited via Fragile X Syndrome Italian Association—Apulian Region, an organization designed to support families and patients with FXS. More specifically, the sample initially consisted of 12 participants, however 2 siblings dropped out after the first few sessions, as their behavior was too challenging for the group dynamics, and they were too young (respectively, 10 and 12 years old) for the mean age of the group. The remaining ten FXS individuals completed all the treatment’s sessions and were then included in the descriptive analysis. The informed consent to participate in the project was obtained from all the parents and the intervention started in March 2020. Table 1 depicts each subject’s gender, age and group mean age at the time of recruitment; group age median (MED) and range are provided.

### 2.2. Implementation of “Corposamente”

A specialized clinical psychologist, with good communication capacities, extensive experience in neuropsychology and a deep knowledge of FXS, carried out the intervention. The psychologist was selected by a group of Apulian families who were looking for a psychosocial intervention for their children. Therefore, as often happens in the clinical practice, participants were not included based on inclusion/exclusion criteria but based on families’ and participants’ needs. Indeed, prior to start of the intervention, the psychologist performed two Skype meetings with parents, to take notice of their requests and propose to them a project that could accommodate every family, while respecting individual differences. More specifically, families were asking for a project: (a) tailored to adolescents’ needs; (b) that could help families and patients to manage FXS symptomatology; and (c) that could encourage participants’ autonomies.

Based on the above, the psychologist designed “Corp-osa-Mente” (CoM), a combined psychosocial–neuropsychological group project that was supposed to dispense one session per month lasting around five hours. Additionally, the psychologist should have carried out cognitive and psychological evaluations before the intervention. However, as of 9 March 2020 the Italian government imposed a national lockdown because of COVID-19, the first sessions were conducted directly online and the first face-to-face meeting was yielded only in July 2020. After the first live session in July 2020, the other nineteen meetings were carried out until September 2022, alternating online small-groups meetings to in person sessions in which all ten participants worked together. Indeed, following the encouraging results that the group obtained working via skype and considering the long distances that some families needed to see the psychologist, the group decided to occasionally use telehealth. 

### 2.3. Treatment Aims Procedures and Settings

The main aims of CoM were to:Increase participants’ knowledge about FXS, its symptomatology and its management.Provide information about adolescence, transition to adulthood and sexuality issues.Help participants to manage their negative emotions, especially anxiety.Promote the development of new autonomies, mostly in the community.Improve the socio-pragmatic skills and the ability to relate with the group of peers.Improve cognitive abilities, with a particular focus on EF.

In order to achieve these objectives, the following methodology was used: 

Psychoeducation aimed to increase patients’ knowledge of FXS. Symptomatology was presented to the group through frontal lessons, written material to read at home and videos that had been shot and edited directly by the psychologist. For instance, through the modification of an episode of “Exploring the human body”, the psychologist coached the participants about FXS etiology and mechanisms of action. Participants were also instructed to identify their own symptomatology by using a symptoms’ checklist (i.e., by providing participants a list of symptoms, participants were required to recognize the ones that they presented). In addition, the psychologist trained the group about associated disorders (i.e., *FXPOI* in women) and about the importance of prevention, by using frontal lessons and providing material to study at home. Additionally, information about adolescence and sexuality issues were provided. The final purpose of patients’ education was to enable the patient to engage in behavior change (i.e., “if I am aware that FXS is associated with learning difficulties, I can improve my abilities by attending cognitive trainings”). Psychoeducation was a part of every session and during the last meeting, the participants shot a new video in which they directly explained to the community what FXS is. This video has been translated to English tongue with the aim to disseminate information about FXS by placing it on the National and International websites of FXS Associations (https://www.youtube.com/watch?v=GFXu2_gPOHE&t=9s, URL accessed on 5 November 2022). 

Cognitive-behavioral techniques based on the model of “Rational emotive behavior therapy (REBT)” [19], which was developed by Albert Ellis in the mid-1950s for the treatment of school-age childhood and adolescent maladjustment. The aim of REBT is to modify dysfunctional behavior and emotions through the modification of irrational thinking, for instance by using the ABC (antecedent–belief–consequence) method. These techniques, although presented to the entire group, were mostly used with higher-functioning participants. Materials were extracted and adapted to the sample from the book “The ABC of Emotions—Age 8–13” [20]. At the end of each section, participants received homework and caregivers were coached to provide them adequate help. 

Behavioral techniques, with the aim to decrease problem behaviors and increase verbal behavior and communication, adaptive skills and social abilities. Among those strategies, the following were preferentially applied:Positive reinforcement, consisting in a reward/praise that was provided in order to encourage positive behaviors.Antecedent-based Interventions, focused on the modification of the environment to reduce the likelihood that something in the environment could trigger an interfering behavior.Extinction, consisting mostly in ignoring dysfunctional behaviors. When one participant exhibited a dysfunctional behavior, the entire group was coached not to pay attention to it.Modelling and Shaping, in order to reward the desired behavior.

Additionally, the psychologist coached caregivers (family members and educators) to use those techniques at home. Checklists and ABC (Antecedent Behavior Consequence) schemes were used during all the intervention. Behavioral strategies were mostly applied with participants with more severe ID and/or with the ones exhibiting challenging behaviors. The following techniques were extracted from “Il manuale ABA-VB-Applied Behavior Analysis and Verbal Behavior: Fondamenti, tecniche e programmi di intervento” [21]:Mindfulness Techniques and physical exercise, in order to help participants to calm down and to manage their negative emotions.Neuropsychological intervention that aimed to train visual attention, different components of EF, perspective memory, receptive, written and expressive language. Material was adapted to each participant’s cognitive level, with the difficulty of the tasks growing session to session. At the end of every session, the psychologist assigned cognitive homework to be completed before the next meeting. Tasks were extracted and adapted by the book “Una palestra per la mente” [22]. Other exercises were directly designed by the psychologist based on her neuropsychological experience.Occupational intervention, aimed to engage participants in meaningful activities of daily life. Activities included tasks of counting money and change, grocery stores and bars simulations, role playing activities to learn to start and maintain conversations with peers, etc. Occupational intervention focused mostly on Socialization and Community living.

The online sessions, lasting around one hour for each small class, were conducted in small groups in which participants were assembled based on their cognitive abilities. The face-to-face meetings took place in different cities around Puglia in order to avoid the same families always having to move to meet the psychologist and to promote (over time) the participants’ abilities to move independently by taking bus or train. The in-person sessions lasted about 5 h with a one-hour lunch break in which the group got used to spending time eating and playing together. In the face-to-face meetings, an educator was involved too, in order to help the psychologist to manage the group. In total, 20 sessions were performed, with the last in September 2022. 

## 3. Results

Treatment effectiveness has been investigated by describing the daily-life behavior at the time of recruitment and at the end of the intervention and by asking participants, their families and educators to fill a semi-structured survey. This information is presented in three different main paragraphs. 

### 3.1. Sample Description in March 2020, at Time of Recruitment

Information about the participants’ ID level and psychiatric comorbidities have been obtained by the most recent medical evaluations reported by local MDs (medical doctors, MDs) (2019–2020) and collected by the psychologist through online clinical interviews with parents. Four (two males, two females) of ten participants exhibited mild ID, two (one male and one female) were diagnosed with moderate ID, while the remaining four with severe ID. All the participants could read and write. None of them exhibited a comorbidity with ASD, as shown by DSM-5 based diagnostic assessments performed by local MDs. Just one boy (with mild ID) was under medication for anxiety. Anxiety symptoms were present in all the participants. Furthermore, one boy exhibited co-occurring ADHD symptomatology (Participant’s Number Code 5, see Table 2) while one girl presented impulse-control disorder symptoms (Participant’s Number Code 2, see Table 2) and one boy depressive ones (Participant’s Number Code 9, see Table 2). No other symptomatology was observed in the sample at time of recruitment. None of the participants attended other group psychological treatments during CoM’s intervention. Seven out of ten participants had a personal assistant/educator that remained the same until the end of the project. Table 2 provides each participant’s ID level and some adaptive functioning information. 

### 3.2. Sample Description in September 2022, at the End of the Intervention

Based on a qualitative analysis, comparing information at time of recruitment and at the end of the intervention (Table 2 and Table 3), it seemed that overall, the entire group exhibited behavioral and adaptive functioning changes at the end of CoM’s intervention. For instance, one girl (Participant’s Number Code 1) got the motivation to attend university, applying some coping strategies against anxiety (“Before every exam, I repeat to myself the memo that I learnt with the psychologist, which is that I have to send anxiety to sleep”). Another girl (Participant’s Number Code 3) learnt some study techniques and got the self-efficacy that allowed her to finish all the bachelor’s exams and to get a bachelor’s degree with a thesis on language impairment in FXS. The girl with impulse-control difficulties (Participant’s Number Code 2), became more capable to recognize triggers to her irrational thinking and to dysfunctional behavior, showing some abilities to make changes in her daily-life to avoid bad consequences (i.e., “when I feel that that I am going to binge, I call a friend of mine to go out”). The boys with more severe ID achieved some daily-life autonomies (i.e., to cook, make the bed, to take a bus alone etc.), exhibiting a greater self-efficacy, self-esteem and self-pride (“I am happy because I can do more things on my own”). All parents reported in their children a general improvement in daily-life autonomies as well as the impression of a greater ability to cope with anxiety.

### 3.3. The Survey

Questionnaires were sent via emails to the families, participants and educators and printed directly by the respondents, with the directive not to specify name and surname. During the last meeting, respondents put their anonymous surveys in a folder, only collected by the psychologist afterwards.

The survey consisted of structured, forced-choice questions plus one open question, with questions focused on the following information: (1) Degree of satisfaction on a five-point Likert scale; (2) Areas of improvement; (3) What respondent liked the most about the intervention; (4) Techniques considered to be the most effective; (5) Family only—the Intervention’s effect on their relationship with the participant; (6) Educators only—what they gained from the Intervention; and (7) Some suggestions.

### 3.4. Descriptive Analysis of the Survey

In total, 27 people responded to the survey, including the 10 participants, 12 parents (seven mothers and five fathers), one sibling and four educators. For each forced-choice question, the percentage of respondents (n_tot = 27) who chose each sub-answer has been calculated. The percentages relative to the sub-groups of respondents (participants, n_ps = ten; family members, n_fm = thirteen; educators, n_edu = four) have been determined too.

#### 3.4.1. Treatment Satisfaction

Treatment satisfaction was scored on a five-point Likert scale, where “5” was “totally satisfied” and “1” was “not at all satisfied”. Figure 1 depicts the overall treatments satisfaction, distinguishing between sub-groups of respondents.

As Figure 1 shows, 74% of respondents were totally satisfied by the intervention, with eight/ten participants and nine/thirteen family members scoring 5. Just one person (one mother) marked “2”.

#### 3.4.2. Areas of Improvement

Respondents completed the following question: “which of the following areas improved after the treatment? You can choose more than one”: (a) domestic skills (i.e., preparing meals alone); (b) social functioning (i.e., going out with friends); (c) autonomies in the work/school environment (i.e., organizing one’s own work, greater management of study materials); (d) receptive skills (i.e., greater ability to understand other people words); (e) expressive skills (i.e., usage of new words; longer and more complex speech); (f) reading-writing skills; (g) executive functions, attention and memory (i.e., increased attention skills, greater ability to memorize information and greater planning abilities); (h) management of emotions/behavior (anxiety, fear, sadness); (i) emotions verbalization; (l) other_specify. Most respondents (74%), with a strong agreement between subgroups, conveyed that they had the impression that participants were more capable to manage emotions and behavior, followed by a global impression of an amelioration in cognitive functions (52%) (Figure 2). Additionally, nine out of thirteen family members observed an amelioration both in receptive and in expressive speech. On the other hand, scores with lower global prevalence were Social Skills and Reading–Writing. The less rated item was Domestic Skills, with just one participant and four/thirteen family members choosing this item (22%, Figure 2). No one scored the item “Other”.

#### 3.4.3. The Most Preferred Aspects of the Intervention

Respondents answered the following question: “which of the following aspects of the intervention did you prefer? You can choose more than one”: (a) in person meetings; (b) online meetings; (c) WhatsApp group (with the other participants, educators and the psychologist); (d) your personal relationship with the psychologist; (e) homework (cognitive and psychological tasks to be performed between sessions); (f) parents’ meetings (online meetings every three sessions with participants); (g) attending conferences and seminars, with the chance to talk in first person; (h) other_specify. Figure 3 exhibits respondents’ preferences.

In total, 93% of the respondents scored the item “In person meetings” as the most preferred element, with all ten participants rating this answer. Furthermore, eight out of ten participants also rated “WhatsApp group” as one of the best Intervention aspects. Additionally, 54% of family members and educators considered the relationship with the psychologist one of the most important ingredients of the treatment. Interestingly, only four/ten participants and two/thirteen family members enjoyed homework. No one scored the item “Other”.

#### 3.4.4. The Techniques Considered the Most Effective

Respondents scored the following forced-choice question: “which of the following intervention’s strategies do you believe to be the most effective? You can rate more than one”: (a) psychoeducation; (b) mindfulness; (c) cognitive techniques (i.e., working on irrational thinking); (d) neuropsychological (cognitive training); (e) behavioural strategies; (f) active listening by the psychologist (counselling); (g) time management (help to organize daily life activities); (h) leisure activities (having lunch together; online games played during sessions, etc.); (i) occupational therapy; (l) sexual education; (m) mindfulness. Occupational therapy was considered the most effective strategy by the 52% of respondents as single group (see Figure 4), with seven/ten participants rating this choice. Just following, 48% of the whole group believed that the cognitive-behavioral strategies were efficacious in treating FXS symptomatology, with the highest number of family members selecting this answer. Notably, 60% of the participants scored “neuropsychological intervention” as frequently as “leisure activities”. Interestingly, all the sub-groups considered behavioral ones the least resultant techniques.

#### 3.4.5. Family Members Only

Thirteen family members answered the following question: “Since the intervention started… You can choose more than one”: (a) I feel more capable to understand the emotions of the participant; (b) I feel more capable to interact with the participant; (c) I feel less distressed only; and (d) All answers. As Figure 5 exhibits, more than half of the family members reported a greater ability to understand the emotions of the person with FXS. “All answers” was at the second highest place. Interestingly, none of the participants reported feeling only less distressed.

#### 3.4.6. Educators Only

The survey was completed by four personal educators, who were additionally asked to rate the following question: “As educator, I enjoy the project because...You can choose more than one”: (a) I know that I can talk with the psychologist for a supervision; (b) I feel more capable to understand the emotion of the person with FXS; (c) I have a deeper knowledge of FXS; and (d) All answers. Figure 6 depicts the percentages of educators responding each answer.

#### 3.4.7. Some Suggestions

Eighteen out of twenty-seven respondents answered the open question “Some suggestions for the psychologist for prosecution and for future interventions”. Here, data were first open coded for themes; thereafter a keyword was associated to each theme. As Figure 7 shows, five compilers (all family members) asked to increase the frequency of the meetings with families; four respondents asked to carry out other meetings and to increase their frequency. At the same level, three/eighteen people asked the psychologist to further focus on social and work functioning and to train other professionals.

## 4. Discussion

This paper presents CoM, a group intervention combining cognitive-behavioral, occupational and neuropsychological techniques that has been carried out for the treatment of ten individuals with FXS.

The intervention started in March 2020 during the COVID-19 global pandemic and lasted until September 2022, providing in total 20 sessions with the alternation of tele-health meetings to in person ones. Both before and at the end of the intervention, the psychologist carried out clinical interviews with FXS individuals and their families, and during the last session handed out a survey about CoM’s efficacy that family members, participants and educators anonymously compiled. Responses to the interview, as well as descriptive analyses about participants’ adaptive behavior prior to and after the intervention are presented in this study.

As emerged from clinical interviews with FXS participants and their parents, the clinical impression was that at the end of the treatment the FXS individuals included in the study were more self-confident, aware of their emotions and of FXS symptomatology and capable to manage daily-life challenges. Everyone achieved new autonomies that were different among participants and specific to each person’s starting skills. Someone learnt to take means of transport, some participants became capable of cooking food and making their own bed and someone else to manage their own money. One girl, after having acquired some cognitive-behavioral strategies to cope with anxiety (i.e., performance anxiety) decided to attend university. The participants all became more able to communicate their ideas and emotions. Everyone developed the need to be more independent and to acquire new skills. Both parents and children reported greater autonomies and less need of support after the intervention. Furthermore, we noticed that other participants’ encouragement and group cohesiveness as well as the psychologist’s conviction and reinforcement that everyone could improve, pushed everyone not to give up, as already demonstrated by previous works on group dynamics, i.e., [23]. In each session, the slogan was “If the others can do it, so can I”, with the entire group really sharing the happiness for each individual success. This observation underlines and confirms the importance of social reinforcement and of the environment in the learning of new abilities [24,25].

Regarding the examination of CoM’s feasibility and efficacy, it is necessary to underline that, as the program started during the pandemic and as families were scattered around the Apulia region, no standardized evaluations have been carried out before and after the intervention. Therefore, no standardized data and systematic analysis can be provided in this study. To curb this limitation, at least partly, at the end of the intervention the psychologist carried out a survey in order to collect qualitative information from participants, families and caretakers.

In summary, the main conclusions of the survey data indicate that CoM was acceptable and appreciated by almost all participants, family members and educators.

Furthermore, family members and educators reported an improvement in the ability of the youths with FXS to manage their emotions. Indeed, as also emerged during clinical interviews, the treated individuals became more aware about their emotions, capable of distinguishing between them (i.e., between anxiety and anger) and to cope with anxiety. Our data is in accordance with previous findings about the efficacy of cognitive-behavioural therapy in the improvement of emotions management both in the non-affected population, i.e., [26] and in people with ID [27,28].

The second highest rated choice was “improvement of executive functions (EF) and other cognitive abilities”. Indeed, during each meeting and between sessions, all participants underwent cognitive training involving exercises for EF, working memory, attention, language and reading/writing. Overall, even though it cannot be proved by standardized data, according to their parents, participants exhibited a qualitative amelioration in cognitive functioning, such as in the ability to make plans (i.e., not making different commitments at the same time), to remember things (i.e., shopping list) and to use language (i.e., having longer conversations with friends). The observation of a general improvement after the neuropsychological training agrees with Hessl and colleagues [29], reporting that cognitive training can improve EF in children and adolescents with FXS.

Interestingly, respondents did not report a great effect on domestic skills. It is possible that, as domestic and personal functioning is already a strength in FXS [12], and as the occupational intervention used by the psychologist did not directly act on home environments, the amelioration of this area was eventually only incidental or secondary.

Furthermore, it is important to notice how social skills remained a major concern after the intervention. A possible explanation is that the improvement in social skills observed in the clinical setting did not always result in real changes in daily life, underlying again the role of environment in symptomatology severity [14].

In order to decide what to include in future interventions, the interview also aimed to detect the favorite elements of CoM. In person meetings was the top-rated choice, highlighting that even though telehealth has been shown to be effective in the treatment of problem behaviors in various populations including FXS [30], patients generally prefer in person visits, mostly if there is the chance to interact with peers. Interestingly, online meetings were less enjoyable for parents than for participants, probably because in person sessions may give parents the chance to have “free-time” or because the use of technological devices represented an initial obstacle for different families.

Regarding the techniques considered to be the most effective by respondents, it emerged that occupational therapy was the most rated option. Indeed, considering the general absence of challenging behaviors, it is probable that our sample benefitted more from all the techniques that could be directly applied during daily-life. These results support previous works indicating that parents of persons with ID choose occupational ones over other therapies because it is generally more practical [31].

Cognitive-behavioral therapy was at the second highest place. We noticed that this option was mostly rated by family members, who were involved in the first person in helping participants to recognize and modify irrational thoughts. Again, this result is in accordance with previous studies highlighting that FXS people can benefit from individual or group cognitive psychotherapy in the management of anxious and depressive symptomatology [32,33].

Additionally, respondents considered psychoeducation one of the most efficacious strategies, confirming the idea that people with ID can reliably report on their own experiences, thoughts and feelings when communication support and clear explanations are provided [34]. Indeed, our participants became averagely capable of recognizing their own symptomatology and correctly explaining to other people what FXS is.

Interestingly, one of the lowest rated choices was “behavioral intervention”. During clinical interviews, some parents reported that they considered behavioral techniques too rigid and imperative, pointing out that these strategies do not completely respect individual differences, in accordance with the work of Leaf and colleagues [35] on major concerns about ABA-based intervention. Among other things, it is possible that as participants did not averagely exhibit challenging behaviors and as none of them presented a comorbidity with ASD, both parents and participants considered behavioral techniques less effective than occupational, cognitive and neuropsychological ones that were better suited to our sample’s features.

Furthermore, parents were interviewed about their feelings and impressions after the intervention, responding to a specific question both in the survey and during clinical interviews. They reported a greater ability to understand their children’s emotions and an ability to interact with them, therefore feeling less distressed. This data is in line with findings documenting increases in parents’ responsive behavior and children’s social interactive behavior after cognitive intervention and psychoeducation, i.e., [36].

Moreover, the four educators who filled the survey responded to a question specifically designed for them, reported a greater confidence with FXS after the intervention. Indeed, the psychologist gave specific advice to the caretakers and oversaw their work, becoming a point of reference for them. This observation underlines the importance of teamwork for the support of people with ID and rare genetic syndromes that are associated with specific cognitive-behavioral phenotypes.

Finally, at the end of the survey, respondents had the chance to give some suggestions to the psychologist. Noticeably, there was a wide demand to carry out a greater number of meetings with parents, underlying that parents of people with ID exhibit both the need to be directly involved in their children’s treatment and to receive first person psychoeducational support. The benefits of parents’ participation to the treatment of children with FXS have already been depicted by Alfieri and coll [37]. In a study in which they demonstrated how cooperative parent-mediated therapy (CPMT) has encouraging results in the treatment of socio-communicative skills in young children with FXS.

Parental involvement and psychoeducation should always be a part of children’s intervention, not only to help parents to manage their children behavior, but also to increase their insights about treatment options. Indeed, in our sample, while considering occupational intervention the most effective technique, parents reported the ability to manage negative emotions as the main area of improvement in their children, which can be considered a result of cognitive-behavioral intervention. A possible explanation is that parents (mostly of adolescents and young adults) usually prefer more practical interventions among other ones, without considering that a greater ability to cope with negative emotions and to manage anxiety may indirectly influence the capacity to live independently. Consequently, as behavioral interventions are associated with better outcomes [15], we believe that educating parents of children with FXS about the differences between types of intervention may be helpful and beneficial for both patients and parents.

In summary, these preliminary data and qualitative analyses encourage cognitive-behavioral and neuropsychological interventions in the treatment of FXS symptomatology. Our exploratory study suggests that group therapy for the management of FXS cognitive-behavioral phenotypes may be a promising approach to continue to pursue, mostly in adolescence when social engagement and daily-life functioning become more demanding.

## 5. Limitations, Future Directions and Conclusions

This study presents the following main limitations: (1) The absence of standardized data prior to and after the intervention (i.e., intellectual quotient -IQ or Adaptive Behavior Composite -ABC scores) which are essential for proper comparisons; (2) The administration of a non-validated survey, even though, as the study was retrospective, our aim was to specifically understand if the techniques that we used were efficacious on participants’ and families’ perspectives; (3) The sample size that could be considered small from a purely statistical point of view, but representative if considering the rarity of the condition. Effectively, it must be pointed out that the treatment was thought for participants with FXS of around the same age and living in the same region; (4) The lack of a secondary comparison group (wait-list control or clearly inactive control), although it must be pointed out that at time of recruitment this project did not have any research purpose. Clearly, the absence of a control group impedes the chance to draw conclusions, but it must be underlined that our main objective was to share our project with both clinical and research community; (5) The inhomogeneity of the sample, indeed males were more than twice the number of females and ID levels were variable. However, this variability again underlines the gap between research and clinical practice and the difficulty to apply rigid methodology in “real life settings”. Future research is warranted based on these preliminary results as a more rigorous research design (i.e., by using RCT—randomized controlled trial design) can address these methodological limitations. Indeed, it is important to exclude that eventual improvements can be due to placebo effects as already discussed by Luu and colleagues in a recent metanalysis aimed at studying the placebo response in clinical trials conducted with children and adults with FXS [38]. Longitudinal studies could also help clinicians and researchers to better understand the reasons for improvements both in drug and in behavioral trials.

In conclusion, while there are several limitations to this study, the initial feasibility and survey results suggest that the combination of occupational, cognitive-behavioral and neuropsychological approaches may improve FXS symptomatology, promoting the achievement of new daily-living autonomies in young adults. This study is also important as it reflects the gap between research and clinical practice, with the hope to encourage other professionals to share their programs with the scientific community, in order to contribute to research on behavioral or psychosocial intervention in the field of FXS.

## Figures and Tables

**Figure 1 brainsci-13-00277-f001:**
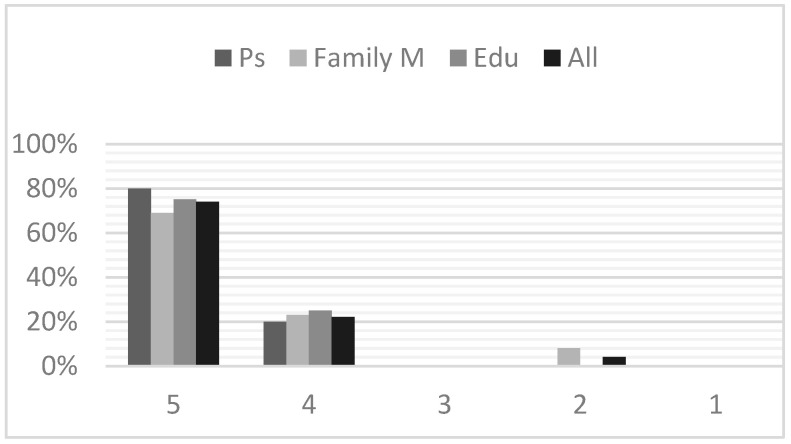
Treatment satisfaction: 5 = totally satisfied; 4 = very satisfied; 3 = satisfied; 2 = partly satisfied; 1 = not at all satisfied. Ps = participants; Family M = family members; Edu = educators.

**Figure 2 brainsci-13-00277-f002:**
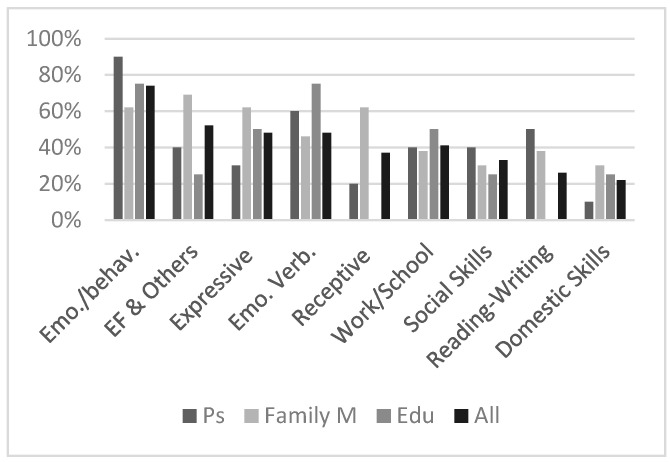
Areas of improvements. Legend: Emo./behav. = emotions and behavioural management; EF and Others = executive functions, attention, memory; expressive = expressive language; receptive = receptive language; emo.verb. = emotions’ verbalization. Ps = participants; Family M = family members; Edu = educators.

**Figure 3 brainsci-13-00277-f003:**
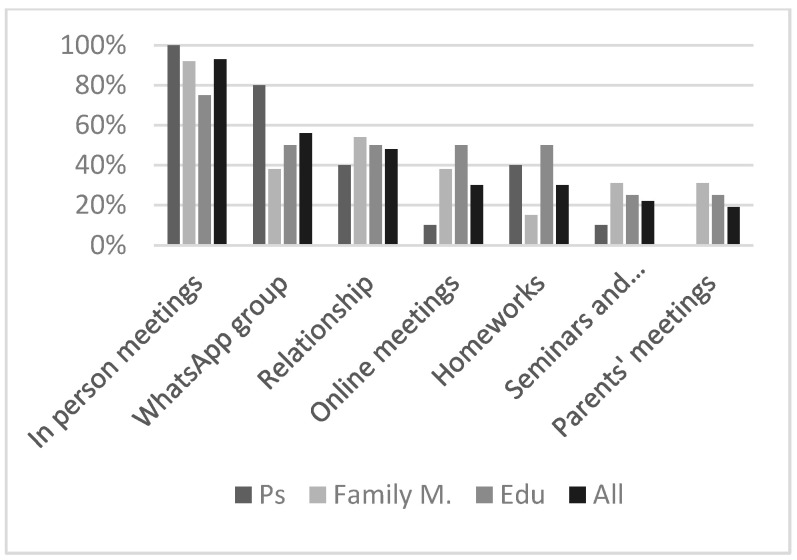
The most favourite elements of the intervention. Relationship = relationship with the psychologist. Ps = participants; Family M = family members; Edu = educators.

**Figure 4 brainsci-13-00277-f004:**
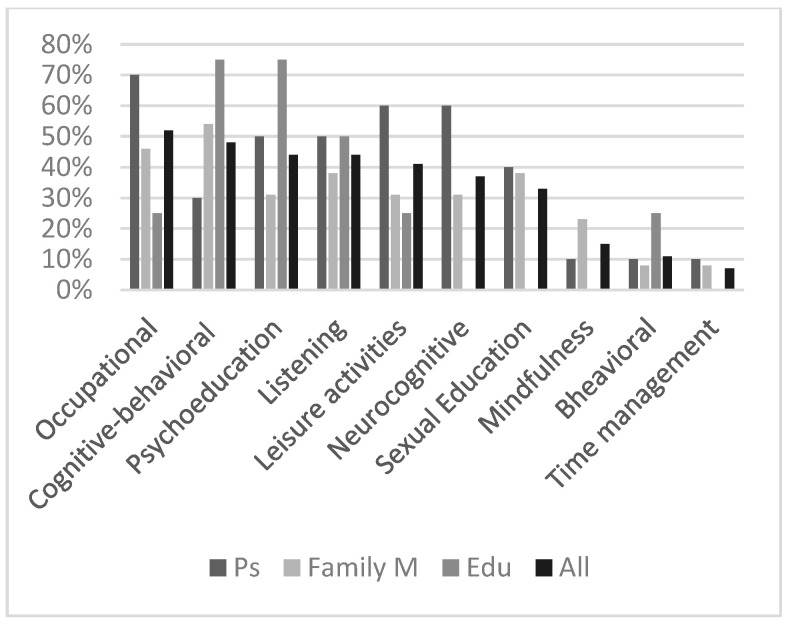
The techniques that the respondent considered the most effective. Occupational = occupational therapy; listening = active listening by the psychologist. Ps = participants; Family M = family members; Edu = educators.

**Figure 5 brainsci-13-00277-f005:**
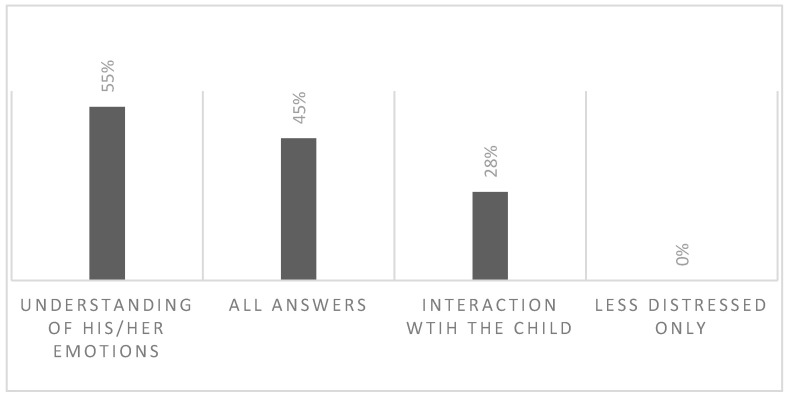
Changes in family members’ feelings after the intervention.

**Figure 6 brainsci-13-00277-f006:**
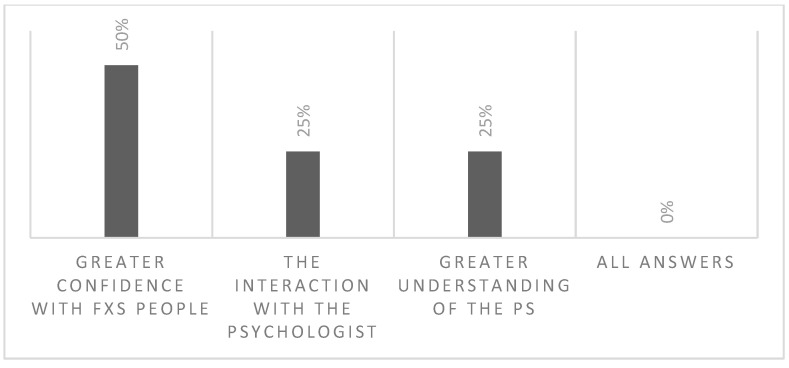
Impressions of the educators after the intervention. PS = participant.

**Figure 7 brainsci-13-00277-f007:**
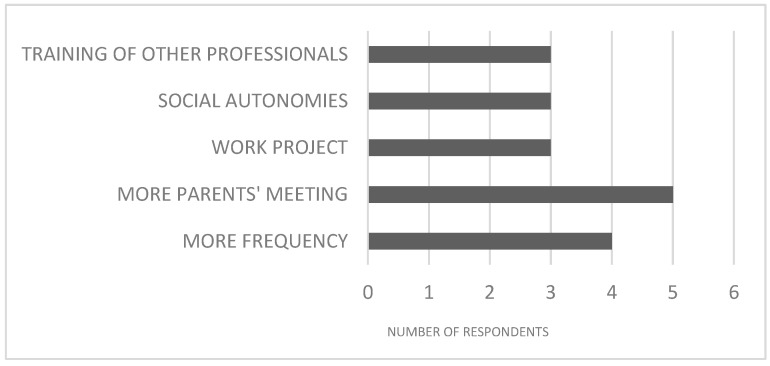
Open question. Suggestions to the psychologist who carried out the intervention. More frequency = of sessions with participants.

**Table 1 brainsci-13-00277-t001:** Participant’s gender and age at recruitment.

Participant’s Number Code	Gender	Age at Recruitment (Years, Months)
1	Female	26.9
2	Female	20.9
3	Female	24.9
4	Male	30.1
5	Male	17.9
6	Male	16.3
7	Male	21.5
8	Male	27.7
9	Male	20.4
10	Male	28.1
Total		M 24.3 SD 4.2 MED 24.9 RANGE 16.3–30.1

Legend: M, mean; SD, standard deviation; MED, median.

**Table 2 brainsci-13-00277-t002:** Participant’s gender, ID level and descriptive information about adaptive skills and behavior at time of recruitment.

Participant’s Number Code	Gender	ID Level	Personal Assistant/ Educator	School/Work	Descriptive Information of Adaptive Behavior
1	Female	Moderate ID	Yes	No occupation	Communication: Can understand complex instructions (receptive skills). Can talk about everyday experiences and clarify to other people what she is thinking or doing (expressive skills). Can pay attention to a 30-min informational talk. Can understand alphabetical order. Can write one-page papers, but with different grammatical errors. Daily living Skills: Very dependent from her family. Can assist in food preparation. Can take care of herself, but she cannot remain alone at home. Socialization: Strong social difficulties and social inhibition. Behavior: Social anxiety traits. She can be very fearful of common situations.
2	Female	Mild ID	Yes	Work in her father’s company	Communication: Can understand complex instructions (receptive skills). Can talk about everyday experiences and clarify to other people what she is thinking or doing (expressive skills). Can pay attention to a 30-min informational talk. Can understand basic sarcasm. Can understand alphabetical order. Can write one-page papers, but with different grammatical errors. Daily living Skills: Can take means of transports alone. Can cook and do the laundry. Can take care of herself and of her brother. Socialization: Severe social difficulties, manipulating behavior with peers and family. Behavior: Can have crises for no clear reason. Can feel helpless. Can destroy her own objects when nervous. This happens at least twice/week and represents a problem for family and peers. On parents’ perspective, these crises represent the main issue.
3	Female	Mild ID	Yes	University student (Individualized Education Program—IEP)	Communication: Can understand complex instructions (receptive skills). Can talk about everyday experiences and clarify to other people what she is thinking or doing (expressive skills). Can pay attention to a 30-min informational talk. Can understand basic sarcasm. Can understand alphabetical order. Can write papers that are three pages or more, but with some grammatical errors. Can read at a sixth-grade level. Daily living skills: Can take means of transports alone. Can assist in food preparation and prepare simple food (i.e., a sandwich) without support. Socialization: Important social difficulties, manipulating behavior with peers. Behavior: Can sometimes worry for no clear reason. Can have crises if she is not the focus of attention (around once/week). This behavior represents a problem for peers, but not for her family.
4	Male	Severe ID	Yes	No occupation	Communication: Receptive and expressive skills strongly impaired, but receptive language is more preserved. Can pay attention to a story for less than 15 min. Can write his name and some basic words. Daily living skills: Can prepare very simple food under supervision. In other activities, total support is required. Socialization: Social drive is present, but with difficulties in understanding social rules. Behavior: Can get fixated on topics. Cannot distinguish between negative emotions.
5	Male	Moderate ID	No	High school student (Individualized Education Program—IEP)	Communication: Can understand complex instructions (receptive skills). Can talk about everyday experiences and clarify to other people what he is thinking or doing (expressive skills). Can pay attention to a 15-min informational talk. Can write one-page papers, but with different grammatical errors. Daily living skills: Can take means of transports alone. Can go out with friends and arrange meetings. Can assist in food preparation. Socialization: Strong social drive. Some difficulties in respecting social rules. Behavior: ADHD traits. Can be irritable and impulsive. Can verbally bully other people or is verbally abusive (at least twice a week). This represents a problem for his parents as this symptom creates obstacles for school and social functioning.
6	Male	Severe ID	Yes	High school student (Individualized Education Program—IEP)	Communication: Receptive skills are more preserved than expressive and written. Can produce basic sentences. Can write his name and other few words. Can pay attention for less than 15 min. Daily living skills: Can prepare very simple food under supervision. Socialization: Social drive is present, but with difficulties in understanding social rules. Behavior: Can get fixated on topics. Cannot distinguish between negative emotions.
7	Male	Severe ID	Yes	No occupation	Communication: Very basic expressive, receptive and written skills. Can produce basic sentences and understand very simple instructions. Can pay attention for around 15 min. Daily living skills: Basic personal and domestic functioning (i.e., can take a shower alone). Can assist in food preparation but needs some support. Can go to the supermarket, buying just one piece at time. Socialization: Strong social drive but shows difficulties in peers’ relationships. Behavior: Can sometimes be verbally abusive with friends and parents (less than once a week), however this symptom does not represent a problem both for parents and friends.
8	Male	Severe ID	Yes	Work in his aunt’s company	Communication: Receptive skills more preserved than expressive and written. Can understand basic instructions and produce basic sentences. Can pay attention for around 15 min. Daily living skills: Can assist in food preparation but needs some support. Can take means of transports alone. Socialization: Strong social drive, but difficulties in peers’ relationships. Behavior: Can sometimes be verbally abusive with friends and parents (less than once/week and only when very distressed). This does not represent a problem for his family and friends.
9	Male	Mild ID	No	No occupation	Communication Can understand complex instructions (receptive skills). Can talk about everyday experiences and clarify to other people what he is thinking or doing (expressive skills), can understand sarcasm and reads at a sixth-grade level. Can produce papers that are three or more pages long. Can pay attention to a 30-min informational talk. Daily living skills: Can cook, can go to the supermarket and can take means of transports alone. Socialization: Important social problems. Social avoidance can be present. Behavior: Can sometimes be sad for no clear reason and feel hopeless (around once/week). This doesn’t seem to represent a problem for his family.
10	Male	Mild ID	No	Work with regular contract in a restaurant	Communication: Can understand complex instructions (receptive skills). Can talk about everyday experiences and clarify to other people what he is thinking or doing (expressive skills), can understand sarcasm and reads at a sixth-grade level. Can produce papers that are three or more pages long. Can pay attention to a 30-min informational talk. Daily living skills: Can cook and take care of himself. Can take means of transport alone. Socialization: Strong social problems. He doesn’t have friends. Behavior: Can sometimes get fixated on topics and situations (less than once/week).

**Table 3 brainsci-13-00277-t003:** Participant’s gender and descriptive information about adaptive skills and behavior in September 2022, at the end of the intervention.

Participant’s Number Code	Gender	School/Work	Descriptive Information of Adaptive Behavior
1	Female	University student (Individualized Education Program—IEP)	Communication: Can understand complex instructions (receptive skills). Can tell about everyday experiences and clarify to other people what she is thinking or doing (expressive skills). Can understand basic irony and write papers that are at least three pages long, even though she can still make some grammar mistakes. Daily living skills: Can go out alone. Can go to the supermarket and cook her own food. Can go to university (in another city) with the support of an educator. Can use money and credit card. Less dependent from her family. Socialization: Still no close friends but can sometimes go to a community center. Behavior: Social anxiety traits are still present, but she is less fearful. Can control her anxiety and approach to novel situations.
2	Female	Work in her father’s company.	Communication: Can understand complex instructions (receptive skills). Can tell about everyday experiences and clarify to other people what she is thinking or doing (expressive skills). Can pay attention to a 30-min informational talk. Can understand basic sarcasm. Can write two-page papers, but with different grammatical errors. Can speak basic English after having attended a course. Daily living skills: Can take means of transports alone. Can manage own money without assistance. Can perform routine household care. Socialization: Still severe social difficulties and dramatic behavior. Behavior: Frequency of crises did not decrease, but now she can recognize triggers to anxiety and behavioral problems. With the help of the psychologist or the educator, can modify irrational thinking and avoid some dysfunctional behaviors. Parents tell to be less distressed as now they know how to manage this behavior.
3	Female	Bachelor’s degree. Masters’ university student. (Individualized Education Program—IEP)	Communication: Can understand complex instructions (receptive skills). Can tell about everyday experiences and clarify to other people what she is thinking or doing (expressive skills). Can makes plans and communicate them to other people. Can pay attention to a 30-min informational talk. Can understand basic sarcasm and be ironic. Can write papers that are three pages or more, but still with some grammatical errors. Can read at a sixth-grade level or more. Daily living skills: Can cook food without assistance. Can take means of transports alone and manage her time independently. Can perform routine household care, with some practical support. She needs help in managing money. Socialization: Some social difficulties mostly in peers’ relationships and theatrical behavior. In a romantic relationship. Behavior: Can still worry for no reason but can accept not to be the focus of attention. Crises are now present once every two weeks, making social inclusion easier.
4	Male	No occupation	Communication: Can now send WhatsApp vocal notes. Daily living skills: Can cook food with minimal support. Can make the bed. Can remember some daily life activities. Socialization: Social drive is present, but he doesn’t have close friends. Behavior: Can now distinguish between some basic emotions (i.e., sadness from anger). Can still get fixated on topics but can now recognize this behavior and cope with it with the help of the educator or the psychologist.
5	Male	Graduated at High school	Communication: Can understand complex instructions (receptive skills). Can tell about everyday experiences and clarify to other people what she is thinking or doing (expressive skills). Can pay attention to a 15-min informational talk. Can write one-page papers, but with different grammatical errors. Daily living skills: Can take means of transports alone. Has a group of friends that meets regularly. Enrolled in a course for driving license. Went on Erasmus with the support of the National Fragile X Syndrome Association. Socialization: Strong social drive, but still difficulties in respecting social rules. Can support his friends and show empathy. Behavior: less pronounced ADHD traits. Less impulsive. Is still sometimes verbally abusive, but frequency is dramatically reduced (less than once/week and only when very distressed). Parents report to be less distressed as the frequency of this behavior is lower and as now, he knows how to cope with it (i.e., by using the mindfulness practices that he learnt during the intervention)
6	Male	High school student. From June to August 2022 got a summer job in a bar.	Communication: Can send WhatsApp vocal notes. Can orally expound basic school material. Can write basic sentences. Daily living skills: Can take the bus alone. Can make his bed, can get a shower alone. Can cook food with some assistance. Socialization: Social drive is present. Can go out with his educator. Behavior: Can become verbally abusive when tired (around once/week) but can now recognize it and cooperate to modify this behavior. This symptom, absent before, appeared during the intervention, and was immediately treated with behavioral techniques. Still not able to distinguish between negative emotions.
7	Male	From June to August 2022 got a summer job in a bar. From September 2022 works in the civil service.	Communication: Still very basic expressive, receptive and written skills. Daily living skills: Can take means of transport with an educator. Can go to the supermarket where buys food with the help of a shopping list. Can plan his daily life with the help of the educator. Socialization: Can go out with and educator who became a friend when his work ended. Behavior: Can now recognize when he is getting nervous and ask for help to avoid verbal offenses.
8	Male	Work in his aunt’s company.	Communication: Expressive and written skills really improved. Receptive skills still more preserved. Friends report that he writes texts and can sustain longer conversations. Daily living skills: Can take means of transports alone. Can make his bed and prepare basilar food without support. He is learning graphic techniques with the support of a video-maker. Socialization: Strong social drive, but still difficulties in peers’ relationships. Behavior: Can recognize when he is getting nervous and ask for help to avoid verbal offenses.
9	Male	No occupation	Communication: Can understand complex instructions (receptive skills). Can tell about everyday experiences and clarify to other people what he is thinking or doing (expressive skills), can understand sarcasm and reads at a sixth-grade level. Can produce papers that are three or more pages long. Can pay attention to a 30-min informational talk. Can write poetries. Daily living skills. Still no occupation, but he is participating in various job competitions. He is attending an English course. Can take means of transports alone. He is getting a drive license. Socialization: Still strong social impairment. Can contact other CoM’s participants and go out with them. Behavior: Can now recognize depressive symptomatology and ask to other members’ group or to the psychologist to help him.
10	Male	Work with regular contract in a restaurant.	Communication: Can understand complex instructions (receptive skills). Can tell about everyday experiences and clarify to other people what he is thinking or doing (expressive skills), can understand sarcasm and reads at a sixth-grade level. Can produce papers that are three or more pages long. Can pay attention to a 30-min informational talk. Daily living skills: Can take means of transports alone. Can prepare food without assistance. Can do the laundry. Socialization: Still strong social problems. Can go to other members’ group hometowns to see them. Decided to go to the swimming pool to meet new people. Behavior: Can now recognize repetitive thought and speech and use cognitive strategies to stop them.

## Data Availability

The original contributions presented in the study are included in the article/Appendix A, further inquiries can be directed to federica.montanaro@opbg.net.

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
