# Peer review of "“Corp-Osa-Mente”, a Combined Psychosocial–Neuropsychological Intervention for Adolescents and Young Adults with Fragile X Syndrome: An Explorative Study"

_brainsci, 2023, doi:10.3390/brainsci13020277_

Round 1
Reviewer 1 Report
Authors present “Corposa-mente”, a combined psychosocial – neuropsychological intervention conducted with a group of ten adolescents/young adults with FXS. In total, 20 sessions were performed, alternating online to face- to-face meetings. In each session, cognitive-behavioural techniques have been used in in combination with neuropsychological and occupational ones, with the aim to improve group’s cognitive abilities and adaptive functioning, as well as participants’ insight about their condition and ability to cope with negative emotions. At the end of the intervention, participants, family members and participants ‘educators anonymously completed a survey that was designed around key areas of improvement as well as treatment satisfaction.
However, as authors noted, there are a lot of limitations. Major revision is needed.
1. The first of all, more participants can be included.
2. Description of the sample must be improved. There are only description data without scoring data, such as ABC scores, or other tests' scores...
3. It is not clear how did authors compare results (from 1st and 2nd section) and how did they conclude that there is improvement? Only by parents' survey? How did they validate this survey?
4. It is necessary to include survey as suppl. data in this article and improve section about surveys results. This section is very confused.
5. It is necessary to clarify what is "Material and methods" and what is "Results"? Just be sure where is appropriate sites to describe sample and results. it is not clear how did the sections look like?
6. Results section: Any score or only description of each included individuals?
7. Many repetitions throughout the whole text. (they repeated a lot of what was already said in the discussion)
8- It is necessary to improve English and a lot minor errors (for example, use italic to write FMR1 gene).
Author Response
We thank the reviewer for his comments. Please find below our answers.
- The first of all, more participants can be included.
We thank you for this suggestion. We are aware that more participants could have been included, however FXS is a rare genetic syndrome, and the study was a clinical project carried out in a small region in the south of Italy. Furthermore, our study is a retrospective study in which we analysed information only afterwards.
- Description of the sample must be improved. There are only description data without scoring data, such as ABC scores, or other tests' scores...
We thank you for this important observation. We are aware of this limitation. Unfortunately, we do not have tests’ scores yet. For this reason, we provide only a descriptive analysis, even though we know it is not enough to draw conclusions.
- It is not clear how did authors compare results (from 1st and 2nd section) and how did they conclude that there is improvement? Only by parents' survey? How did they validate this survey?
Thanks for this consideration. We compared daily-life functioning at time of recruitment (March 2020) with the one at the end of the intervention, September 2022. Information about participants’ autonomies were collected during clinical interviews with parents and participants. In other words, we speculated an improvement based on parents’, patients’ and educators’ reports and on the psychologist’s clinical judgement. Data obtained with the clinical interviews were supplemented with from the survey. We did not use a validated survey, but we constructed a semi-structured interview based on our objectives.
- It is necessary to include survey as suppl. data in this article and improve section about surveys results. This section is very confused.
Thanks for the suggestion. We included the survey as suppl. material. Furthermore, we adjusted the section about surveys results.
- It is necessary to clarify what is "Material and methods" and what is "Results"? Just be sure where is appropriate sites to describe sample and results. it is not clear how did the sections look like?
Thank you for the observation. After your suggestion, we modified the sections and we put the descriptive analysis of the sample in one main paragraph, in order to make the different sections clearer.
- Results section: Any score or only description of each included individuals?
Thank you for your question. It is a descriptive analysis, unfortunately without tests’ scores.
- Many repetitions throughout the whole text. (they repeated a lot of what was already said in the discussion).
Thank you for the observation. We revised the paper, deleting repetitions.
8- It is necessary to improve English and a lot minor errors (for example, use italic to write FMR1 gene).
Thank you for your consideration. We adjusted the paper and used italic for FMR1 gene.
Reviewer 2 Report
Empirical research on the effectiveness of behavioral treatments among patients with FXS is still lacking. In study, “Corposamente”, a combined psychosocial – neuropsychological intervention was conducted with a group of ten adolescents/young adults with FXS. In total, 20 sessions were performed, alternating online to face-to-face meetings. In each session, cognitive-behavioral techniques have been used in in combination with neuropsychological and occupational ones, with the aim to improve group’s cognitive abilities and adaptive functioning, as well as participants’ insight about their condition and ability to cope with negative emotions. At the end of the intervention, participants, family members and participants ‘educators anonymously completed a survey that was designed around key areas of improvement as well as treatment satisfaction. Survey’s results indicated that participants proved mostly in their ability to cope with negative emotions and that occupational intervention was considered the most effective technique. The authors of this pilot study suggests that group therapy to the management of FXS cognitive-behavioral phenotype may be a promising approach to continue to pursue, mostly in adolescence when the environmental demands increase.
This is an interesting pilot work aimed to contribute to a gap in psychosocial studies of adolescents and young adults individuals with FXS. The authors are praised for their effort despite the challenges disclosed.
Major suggestions
Lines 139-153, the authors are praised for being open about the study’s methodological issues.
Lines 195-197
“Cognitive-behavioural techniques based on the model of “Rational emotive behaviour therapy (REBT)” [17], which has been developed by Albert Ellis in the mid-1950s for the treatment of school-age childhood and adolescent maladjustment.”
How is this approach going to be replicated by others? Did the authors standardize this approach? and if yes, what data they can present to support it of relevance for this study.
Lines 204-206
“Applied behaviour analysis (ABA) techniques, with the aim to decrease problem be-204 haviours and increase verbal behaviour and communication, adaptive skills and social 205 abilities.”
So, the study combined Cognitive-behavioural techniques and ABA? This is not a major weakness but needs a more cohesive description.
Results
Lack of application of standardized measures (such as Aberrant Behavior Checklist-fragile Version) that would allow readers to understand/interpret the findings. There are multiple figures (8) and readers would benefit form a ‘summary figure’. The authors are also encouraged to build a case report approach to combine those cases that are similar and present as such, which would sort of subtype their sample.
Methods
Lines 112-113
The intervention has been performed with ten FXS participants (M: F = 7:3) with 112 FMR1 full mutation, as determined by DNA testing.
What the DNA testing method was specifically used?
Discussion lines 474-477
“the second highest rated choice was “improvement of EF and other cognitive abili-474 ties”. Indeed, during each meeting and between sessions, all participants underwent a 475 cognitive training involving exercises for EF, working memory, attention, language and 476 reading/writing. Overall, participants exhibited an amelioration in cognitive functioning, such as in the ability to make plans.”
Did the study apply any standardized neuropsychological cognitive, adaptive, EF measures to show the above reported and other findings?
Minor suggestions
Introduction
Lines 39-40
the FMR1 (Fragile X Mental Retardation 1, today renamed Fragile X messenger ribonucleoprotein 1)
FMR1 ought to be in italic throughout the body text. And renamed Fragile X messenger ribonucleoprotein 1.
Line 42
FMRP protein,
please spell out FMRP initially.
Line 76
“Despite to the clear knowledge of..”
Line 1322
“Indeed, prior to start the intervention, the..”
And other examples, together, manuscript would benefit from proofread of a native English speaker.
Lines 89-93
Consequently, while the experts who are part of the National Fragile X Foundation’s Fragile X Clinical & Research Consortium (FXCRC) offer recommendations for medications in FXS, there are no systematic guidelines for behavioral treatments, reason for which psychologists generally still rely on their clinical experience.
The authors are encouraged to include published literature/citations from members of FXCRA that include large database evidence of underuse of behavioral services, including ABA, social training, tutoring, and vocational training, in individuals with FXS co-morbid with ASD https://doi.org/10.1542/peds.2016-1159F
Line 432
...of ten FXS individuals"
..of ten individuals with FXS".
Author Response
We thank the reviewer for his major and minor suggestions. Please find below our answers.
Major suggestions
Lines 139-153, the authors are praised for being open about the study’s methodological issues.
Thanks for your consideration. We modified the paragraph, making it more formal.
Lines 195-197
“Cognitive-behavioural techniques based on the model of “Rational emotive behaviour therapy (REBT)” [17], which has been developed by Albert Ellis in the mid-1950s for the treatment of school-age childhood and adolescent maladjustment.”
How is this approach going to be replicated by others? Did the authors standardize this approach? and if yes, what data they can present to support it of relevance for this study.
Thank you for your observation. There is a robust empirical evidence about the efficacy of Rational emotive behaviour therapy (see David, D., Cotet, C., Matu, S., Mogoase, C., & Stefan, S. (2018). 50 years of rational-emotive and cognitive-behavioral therapy: A systematic review and meta-analysis. Journal of clinical psychology, 74(3), 304–318. https://doi.org/10.1002/jclp.22514). Data have been replicated by different studies and applied in different domains, like clinical psychology, education, etc. REBT techniques have been used in this study to modify irrational thinking that precedes the dysfunctional emotions exhibited by people, in this case with FXS.
Lines 204-206
“Applied behaviour analysis (ABA) techniques, with the aim to decrease problem be-204 haviours and increase verbal behaviour and communication, adaptive skills and social 205 abilities.”
So, the study combined Cognitive-behavioural techniques and ABA? This is not a major weakness but needs a more cohesive description.
Thank you for the suggestion. We did not use the ABA methodology as one of the main strategies, but only some behavioural techniques. We modified the section.
Results
Lack of application of standardized measures (such as Aberrant Behavior Checklist-fragile Version) that would allow readers to understand/interpret the findings. There are multiple figures (8) and readers would benefit form a ‘summary figure’. The authors are also encouraged to build a case report approach to combine those cases that are similar and present as such, which would sort of subtype their sample.
Thank you for the suggestion. We are aware that the absence of standardized measures is one of the main limitations of the study. We did not use a “summary figure” as figures refers to different questions. However, after your suggestion, we decided to change the section and use less figures. Finally, we modified the section about results in order to follow a descriptive approach.
Methods
Lines 112-113
The intervention has been performed with ten FXS participants (M: F = 7:3) with 112 FMR1 full mutation, as determined by DNA testing.
What the DNA testing method was specifically used?
Thank you for the question. We added it in the paper. The test is: Fragile X CGG repeat analysis Test.
Discussion lines 474-477
“the second highest rated choice was “improvement of EF and other cognitive abili-474 ties”. Indeed, during each meeting and between sessions, all participants underwent a 475 cognitive training involving exercises for EF, working memory, attention, language and 476 reading/writing. Overall, participants exhibited an amelioration in cognitive functioning, such as in the ability to make plans.”
Did the study apply any standardized neuropsychological cognitive, adaptive, EF measures to show the above reported and other findings?
We did not use standardized measures. We administered some standardized tests, but we did not have the possibility to make comparisons between pre and post intervention. Improvements have been speculated based on clinical judgment and parents’ reports.
Minor suggestions
Introduction
Lines 39-40
the FMR1 (Fragile X Mental Retardation 1, today renamed Fragile X messenger ribonucleoprotein 1)
FMR1 ought to be in italic throughout the body text. And renamed Fragile X messenger ribonucleoprotein 1.
Thank you for your suggestion. We put FMR1 in italic in all the paper.
Line 42
FMRP protein,
please spell out FMRP initially.
Thank you. We adjusted it.
Line 76
“Despite to the clear knowledge of..”
Thank you for your suggestion. We adjusted it.
Line 1322
“Indeed, prior to start the intervention, the..”
Thank you for your suggestion. We adjusted it.
And other examples, together, manuscript would benefit from proofread of a native English speaker.
According to your suggestions, we revised our English.
Lines 89-93
Consequently, while the experts who are part of the National Fragile X Foundation’s Fragile X Clinical & Research Consortium (FXCRC) offer recommendations for medications in FXS, there are no systematic guidelines for behavioral treatments, reason for which psychologists generally still rely on their clinical experience.
The authors are encouraged to include published literature/citations from members of FXCRA that include large database evidence of underuse of behavioral services, including ABA, social training, tutoring, and vocational training, in individuals with FXS co-morbid with ASD https://doi.org/10.1542/peds.2016-1159F
We modified that section, added more citations and examples.
Line 432
...of ten FXS individuals"
..of ten individuals with FXS".
Thanks for the encouragement to change. We modified the text in the paper.
Reviewer 3 Report
As the authors tell this project was not intended to be a research and this is the main issue. There were no base-line assessments of the adaptive nor cognitive skills and no systematic follow-up method. Data were collected from previous medical reports. There were four with mild ID, two with moderate ID and two with severe ID. At which age were the study members tested and with which test methods? The severity, the level of ID may change from childhood to adulthood. The adaptive skills described in the tables are not systemically written. The eight study member with severe ID had learned to write?
I do believe that the study members benefited from this rehabilitation and it is most important to offer occupational therapy to people with ID.
Author Response
We thank the reviewer for the considerations. We are aware of the strong limitations of the study. Please find below our answers.
There were no base-line assessments of the adaptive nor cognitive skills and no systematic follow-up method. Data were collected from previous medical reports. There were four with mild ID, two with moderate ID and two with severe ID. At which age were the study members tested and with which test methods?
Thanks for your question. We did not administer standardized test when we started because of the Pandemic, therefore also the follow-up has been difficult. Clinical information were collected by medical reports. All the ten individuals had recent medical evaluations (2019-2020). After your suggestion, we specified it in the paper: Information about participants’ ID level and psychiatric comorbidities have been obtained by the most recent medical reports (2019-2020) and collected by the psychologist throw online clinical interviews with parents.
The adaptive skills described in the tables are not systemically written.
Thank you for the advice. We modified the tables, describing in a more systematic manner the adaptive skills.
The eight-study member with severe ID had learned to write?
Thank you for the question. They all learnt to write. According to your observation, we specified it in the paper.
Reviewer 4 Report
The article is well written, but I am afraid that I have some significant concerns. There are only 10 subjects, and this is not justified by the use of idiographic research designs such as suggested by Hersen and Barlow. There is really no experimental design; it is just a clinical trial. The treatment methodology is not well enough explained to enable replication. The dependent measures did not seem to have been specifically selected as targets, and there are many concerns about the utility of satisfaction data.
I'd welcome a major rewrite that focused on the details of the treatment. Any such rewrite should be substantially reduced in volume. This is just a clinical trial, and you don't want readers to attribute more to this than is warranted. I'd introduce the treatment approach and briefly discuss outcomes, and then call for an empirical evaluation of the treatment package.
Author Response
We thank the reviewer for his comments. Please find below our answers
The article is well written, but I am afraid that I have some significant concerns. There are only 10 subjects, and this is not justified by the use of idiographic research designs such as suggested by Hersen and Barlow. There is really no experimental design; it is just a clinical trial. The treatment methodology is not well enough explained to enable replication. The dependent measures did not seem to have been specifically selected as targets, and there are many concerns about the utility of satisfaction data.
I'd welcome a major rewrite that focused on the details of the treatment. Any such rewrite should be substantially reduced in volume. This is just a clinical trial, and you don't want readers to attribute more to this than is warranted. I'd introduce the treatment approach and briefly discuss outcomes, and then call for an empirical evaluation of the treatment package.
Thank you for pointing this out. We agree with your concerns however our research is a retrospective study and data were analysed only afterwards. Ten subjects are not that few if we consider that FXS is a rare genetic syndrome and that the treatment was carried out with ten adolescents/young adults scattered around Puglia, a small region in the south of Italy, during the Pandemic. As we specified in the section “limitations”, the absence of a control group obstacles the chance to draw conclusions, but our main objective was to share with the community our project. In the introduction, we anticipated that strong methodological limitations were present in the study in order to advise readers not to attribute more than warranted.
Regarding methodology, we listed the sources (i.e. Books) that we used. We agree with you that we could be more specific, therefore we modified the section.
Thank you also for the observation about satisfaction data. We decided to use this question in order to understand if parents and participants were satisfied about the intervention, as we wanted to continue to project, carrying out standardized evaluations and pursuing new objectives. Finally, accordingly to your advice, we reduced the volume of our paper, using a descriptive approach rather than a idiographic research design.
Round 2
Reviewer 1 Report
The revised article is improved. However, there are still a lot of concerns.
Major: I understand that FXS is rare disorders, but for such a study more patients have to be included. It means that authors need more time for such a study. In addition, the survey is not validated. There is no scores, and no data about IQ scores, too. Results are not clearly presented.
My suggestion is that authors can prepare these results and presented them as case report/s. Moreover, they can choose one the most important case and present that patient. Such article (case report) can be interesting. According to my opinion there are no appropriate data to present descriptive results as original work.
Minor:
1. Line 50-51: Please, improve this sentence: ..."the triplet has 50 between 5 and 45 repeats" in order to make it clearer.
2. Premutation instead of pre-mutation in the whole text.
3. Line 108: Which DNA test did you use? Please, specify PCR or SB method?
Author Response
Reviewer 1
Comments and Suggestions for Authors
The revised article is improved. However, there are still a lot of concerns.
Major: I understand that FXS is rare disorders, but for such a study more patients have to be included. It means that authors need more time for such a study. In addition, the survey is not validated. There is no scores, and no data about IQ scores, too. Results are not clearly presented.
My suggestion is that authors can prepare these results and presented them as case report/s. Moreover, they can choose one the most important case and present that patient. Such article (case report) can be interesting. According to my opinion there are no appropriate data to present descriptive results as original work.
Thank you for giving us the opportunity to submit a revised draft of our manuscript.We appreciate the time and effort that you have dedicated to providing your valuable feedback on our manuscript. Taking into account your comments, we really understand your concerns, however we believe that the final decision should be taken by the editor. We want to point out that after your suggestion, we specified in the introduction that this is simply a descriptive study and we underlined that we don’t have standardized results yet. Furthermore, we revised the paper, making it more like a descriptive study than to an original work. Finally, we changed the title from “pilot study” to “exploratory study” In order to avoid any misunderstanding.
Minor:
- Line 50-51: Please, improve this sentence: ..."the triplet has 50 between 5 and 45 repeats" to make it clearer.
Thank you. We changed the sentence into “CGG repeat number is between 5 and 45”.
- Premutation instead of pre-mutation in the whole text.
Thank you. We modified it.
- Line 108: Which DNA test did you use? Please, specify PCR or SB method?
Thank you for asking. We used methylation PCR method. After your suggestion, we specified it in the paper.
Reviewer 2 Report
Abstract
1) The FMR1 gene should be labeled as an italic…
Keywords: Fragile X Syndrome; FMR1 gene;
Line 27, “
Our pilot study suggests that group therapy..
2) The authors changed the title to ‘exploratory’ in the tittle, but not in the Abstract. There appears to be an inadequate attention to details. This is another example of it.
3) Also, the abstract is lacking an line of the sample description..for example, 10 patients with FXS, non-ASD…which does not help if this gets published to be actually read by a potential reader. The abstract needs more work to make it more concise, to the point.
Lines 41-42 .. “the FMR1 gene that encodes for the FMRP (Fragile X Mental Retardation Protein)”
4) FMRP now stands for Fragile X messenger ribonucleoprotein, and is labeled non-italic per the convention. That should be corrected in Introduction and throughout. Only the FMR1 gene is labeled as an italic.
Lines 114-115
2.1. Participants 113
“The intervention has been performed with ten FXS participants (M: F = 7:3) with FMR1 full mutation, as determined by DNA testing. (Fragile X CGG repeat analysis Test).”
5) ..ten participants with FXS..(this was already recommended in R1 by this reviewer, but apparently was not corrected throughout the body text) FXS …and FMR1 full mutation means the same. Suggest to avoid such repetition in that close space in one sentence.
FXS is a clinical term and FMR1 full mutation is a genetic term that means the same. Pick one depending on the meaning of your sentence.
6) The above are minor but important issues. A remaining major issue is a lack of an adequate description of the Fragile X CGG repeat analysis Test.
Lacking in particular is whether the test measures the methylation status of the gene, or only CGG repeats expansion level. If the latter is the case, that would be a major limitation that should be also reported under the study already significant limitations.
Line 262, …”and collected by the psychologist throw online clinical interviews with parents.
7) It seems that the authors wanted to say …through online..Needs a good proofread by an English native speaker.
8) Line 264, None of them 264 exhibited a comorbidity with ASD.
If true, mention it in the abstract/body text as helps describe the patients profile. Describe a method used, if not done, like DSM-5, clinician based (PhD or MD?, relevant to a level of training, experience in dealing with ASD)
9) In Tables,…” can become verbally abusive..” try to be more specific for the problem behavior, like a frequency (i.e., once a week, once a month) and whether that is considered a problem/issue by the family. Please modify it throughout the Tables. That would help a bit offset the major methodological limitations.
‘Can become..” seems to imply no behavioral issue, but please check it and document as the above suggested.
For skills, “Communication: Can express his own ideas..” try to be more specific, if possible.
Lines 630, 631 Limitations
“Future research is warranted based on these preliminary results as a 630 more rigorous research design (i.e.., by using RCT – randomized controlled trial design) can address these methodological limitations.”
10) Needs reference(s). For example, to mention a potential placebo effect already published in clinical trials studies in FXS (10.3390/brainsci10090629
Author Response
Thank you for giving us the opportunity to submit a revised draft of our manuscript.We appreciate the time and effort that you have dedicated to providing your valuable feedback on our manuscript.We are grateful for the insightful comments on our paper. We have been able to incorporate changes to reflect most of the suggestions provided.
Please see below our response:
Comments and Suggestions for Authors
Abstract
1) The FMR1 gene should be labeled as an italic…
Keywords: Fragile X Syndrome; FMR1 gene;
Thank you for the comment. We modified it.
Line 27, “
Our pilot study suggests that group therapy..
We really thank you for the observation. We changed “pilot” with “exploratory” in all the paper.
2) The authors changed the title to ‘exploratory’ in the tittle, but not in the Abstract. There appears to be an inadequate attention to details. This is another example of it.
Thank you for pointing this out. It was an oversight. We modified it.
3) Also, the abstract is lacking an line of the sample description..for example, 10 patients with FXS, non-ASD…which does not help if this gets published to be actually read by a potential reader. The abstract needs more work to make it more concise, to the point.
Thank you for the suggestion. We modified the sentence, adding some information: “ten adolescents/young adults with FXS, non-ASD and without significant behavioral problems”. We shortened the abstract too.
Lines 41-42 .. “The FMR1 gene that encodes for the FMRP (Fragile X Mental Retardation Protein)”
4) FMRP now stands for Fragile X messenger ribonucleoprotein, and is labeled non-italic per the convention. That should be corrected in Introduction and throughout. Only the FMR1 gene is labeled as an italic.
We really thank you for this consideration. We corrected the paper.
Lines 114-115
2.1. Participants 113
Thank you. We corrected it.
“The intervention has been performed with ten FXS participants (M: F = 7:3) with FMR1 full mutation, as determined by DNA testing. (Fragile X CGG repeat analysis Test).
5) ..ten participants with FXS..(this was already recommended in R1 by this reviewer, but apparently was not corrected throughout the body text) FXS …and FMR1 full mutation means the same. Suggest to avoid such repetition in that close space in one sentence.
FXS is a clinical term and FMR1 full mutation is a genetic term that means the same. Pick one depending on the meaning of your sentence.
Thank you for this comment. Indeed, it is a typo that should be avoided. We modified it.
6) The above are minor but important issues. A remaining major issue is a lack of an adequate description of the Fragile X CGG repeat analysis Test.
Lacking in particular is whether the test measures the methylation status of the gene, or only CGG repeats expansion level. If the latter is the case, that would be a major limitation that should be also reported under the study already significant limitations.
Thank you for the comment. All the participants performed the methylation PCR method test. We have specified it in the study now.
Line 262, …”and collected by the psychologist throw online clinical interviews with parents.
7) It seems that the authors wanted to say …through online..Needs a good proofread by an English native speaker.
Thanks for your suggestion. We modified it and adjusted our English in the paper.
8) Line 264, None of them 264 exhibited a comorbidity with ASD.
If true, mention it in the abstract/body text as helps describe the patients profile. Describe a method used, if not done, like DSM-5, clinician based (PhD or MD?, relevant to a level of training, experience in dealing with ASD).
Thank you for asking to specify this information. We modified the paragraph as follows: Information about participants’ ID level and psychiatric comorbidities have been obtained by the most recent medical evaluations reported by local MDs (2019-2020) and collected by the psychologist through online clinical interviews with parents. Four (2 males, 2 females) of ten participants exhibited mild ID, two (1 male and 1 female) were diagnosed with moderate ID, while the remaining four with severe ID. All the participants could read and write. None of them exhibited a comorbidity with ASD, as showed by clinical assessments performed by local MDs.
9) In Tables,…” can become verbally abusive..” try to be more specific for the problem behavior, like a frequency (i.e., once a week, once a month) and whether that is considered a problem/issue by the family. Please modify it throughout the Tables. That would help a bit offset the major methodological limitations.
Thank you for the suggestion. We have been more specific about frequency both before and after the intervention. We also specified if it was an issue for families and/or friends.
‘Can become..” seems to imply no behavioral issue, but please check it and document as the above suggested.
For skills, “Communication: Can express his own ideas..” try to be more specific, if possible.
After your suggestion, we have been more specific in both the tables. Thank you.
Lines 630, 631 Limitations
“Future research is warranted based on these preliminary results as a 630 more rigorous research design (i.e.., by using RCT – randomized controlled trial design) can address these methodological limitations.”
10) Needs reference(s). For example, to mention a potential placebo effect already published in clinical trials studies in FXS (10.3390/brainsci10090629
Thank you for this important suggestion. After your advice, we adjusted the paper: Future research is warranted based on these preliminary results as a more rigorous research design (i.e., by using RCT – randomized controlled trial design) can address these methodological limitations. Indeed, it is important to exclude that eventual improvements can be due to placebo effects as already discussed by Luu and colleagues in a recent metanalysis aiming to study the placebo response in clinical trials conducted with children and adults with FXS. Longitudinal studies could also help clinicians and researchers to better understand reasons of improvements both in drug and in behavioral trials.
Reviewer 3 Report
This is a story or description of rehabilitation project not a scientific research.
Medical reports were analyzed but no neuropsychological reports. It was not told at which age intelligence tests were performed.
Author Response
Thank you for giving us the opportunity to submit a revised draft of our manuscript.We appreciate the time and effort that you have dedicated to providing your valuable feedback on our manuscript.We are grateful for the insightful comments on our paper. Please find below our response
Comments and Suggestions for Authors
This is a story or description of rehabilitation project not a scientific research.
Medical reports were analyzed but no neuropsychological reports. It was not told at which age intelligence tests were performed.
Thank you for your comment. After your review, we better clarified that our work is just an exploratory study. Intelligence tests were performed between 2019 and 2020. We specified it in the paper. Thank you for the important points that you raised. We hope you will appreciate the modified version of the paper.
Reviewer 4 Report
Thank you for your responses.
Author Response
Thank you for giving us the opportunity to submit a revised draft of our manuscript.We appreciate the time and effort that you have dedicated to providing your valuable feedback on our manuscript.We are grateful for the insightful comments on our paper and we hope you will appreciate the revised version of our work.